biochemistry

glucosyltransferase, glycyrrhetinic acid, glucosylation, *Glycyrrhiza uralensis*

**Authors for correspondence:**
Zhubo Dai
e-mail: dai_zb@tib.cas.cn
Dan Jiang
e-mail: jiangdan1027@163.com
Chunsheng Liu
e-mail: max_liucs@263.net

# GuUGT, a glycosyltransferase from *Glycyrrhiza uralensis*, exhibits glycyrrhetinic acid 3- and 30-O-glycosylation

Ying Huang[1], Da Li[1], Jinhe Wang[2], Yi Cai[2], Zhubo Dai[2], Dan Jiang[1] and Chunsheng Liu[1]

[1]School of Chinese Materia Medica, Beijing University of Chinese Medicine, Beijing 100102, People's Republic of China
[2]Key Laboratory of Systems Microbial Biotechnology, Tianjin Institute of Industrial Biotechnology, Chinese Academy of Sciences, Tianjin, People's Republic of China

YH, 0000-0002-9870-7180; ZD, 0000-0002-2704-4922; DJ, 0000-0001-5146-3812; CL, 0000-0001-8514-8925

*Glycyrrhiza uralensis* is a well-known herbal medicine that contains triterpenoid saponins as the predominant bioactive components, and these compounds include glycyrrhetinic acid (GA)-glycoside derivatives. Although two genes encoding UDP-glycosyltransferases (UGTs) that glycosylate these derivates have been functionally characterized in *G. uralensis*, the mechanisms of glycosylation by other UGTs remain unknown. Based on the available transcriptome data, we isolated a UGT with expression in the roots of *G. uralensis*. This UGT gene possibly encodes a glucosyltransferase that glycosylates GA derivatives at the 3-OH site. Biochemical analyses revealed that the recombinant UGT enzyme could transfer a glucosyl moiety to the free 3-OH or 30-COOH groups of GA. Furthermore, engineered yeast harbouring genes involved in the biosynthetic pathway for GA-glycoside derivates produced GA-3-*O*-β-D-glucoside, implying that the enzyme has GA 3-O-glucosyltransferase activity *in vivo*. Our results could provide a frame for understand the function of the UGT gene family, and also is important for further studies of triterpenoids biosynthesis in *G. uralensis*.

## 1. Introduction

*Glycyrrhiza uralensis* (Chinese liquorice) is a well-known traditional Chinese herbal medicine recorded in the pharmacopoeias of many Asian countries [1]. In particular, its roots are widely used to treat coughs, influenza and liver damage and for the purposes of detoxification in oriental clinical practice [2–5]. Biological studies on liquorice extracts have revealed their antioxidant,

**Figure 1.** The biosynthetic pathways of glycyrrhetinic acid and its glycosylated derivatives. GgbAs, β-amyrin synthetase; CPR, cytochrome P450 reductase; UGTs, UDP-glucuronosyltransferases; Glc, glucose; Glca, glucuronic acid.

anti-inflammatory, antiviral, cytotoxic, antidiabetic and cholinergic effects [6–9]. Triterpenoid saponins represent the predominant bioactive components in *G. uralensis*, and these compounds protect the plant from injury by insects and microbes and improve its stress resistance under poor environmental conditions. Extracts from liquorice primarily include glycyrrhetinic acid (GA) and glycyrrhizin. Notably, glycyrrhizin is more commonly used than GA, as it is a popular antihepatitis drug and is also sweeter to the taste than GA [10]. Several other triterpenoid saponins extracted from *G. uralensis* also exhibit potent biological activity including antibacterial, antiproliferative, anti-H1N1, anti-cancer and anti-HIV properties [11]. These components, such as 18β-GA-30-*O*-β-D-glucoside and uralsaponins M-Y, have been considered for exploitation for the development of new drugs.

The structure of triterpenoid saponins consists of an oleanane-type triterpene skeleton with various sugar moieties (glucose, glucuronic acid, galactose, arabinose and xylose) attached to the hydroxyl and/ or carboxyl groups and even other nucleophilic groups (-SH and -NH$_2$) [12]. 18β-GA is one of the most important oleanane-type triterpenoid sapogenins in *G. uralensis*, and most of the genes involved in its biosynthetic pathway have been successfully cloned and characterized, including β-amyrin synthetase (bAS) [13], cytochrome P450 monooxygenase 88D6 (CYP88D6) [14] and CYP72A154 [15] (figure 1). The free C-3 OH and C-30 COOH groups of GA are readily attached to sugar moieties through β-glycosidic linkages that are formed by glycosyltransferases (UGTs).

Currently, four UGTs from *G. uralensis* have been functionally characterized and shown to naturally synthesize 18β-GA glycoside derivates. GuUGAT catalyses a continuous two-step glucuronosylation reaction at C3-OH of GA to synthesize glycyrrhizic acid [16], whereas UGT73P12 catalyses the second glucuronosylation reaction to biosynthesize glycyrrhizin but has no catalytic activity for GA [17]. Prof. Ye Min's group demonstrated that UGT73F17 selectively catalyses esterification at C30-COOH of GA to synthesize liquorice-saponin A3 and 18β-GA-30-*O*-β-D-glucoside, and GuUGT73F15 can transfer a glucosyl or glucuronyl moiety to the free C3 hydroxyl groups of GA [18,19]. However, these two UGTs comprise a very small fraction of the abundance of UGTs present in the genome of *G. uralensis*. Three UGTs from another plant (*Barbarea vulgaris*, UGT73C11) and a microbe (YjiC1 and Bs-YjiC) have also been demonstrated to glycosylate the free 3-OH of GA with glucose as the sugar moiety, and Bs-YjiC can also transfer a glucosyl moiety to the free 30-COOH of GA and glycyrrhizic acid [20–22]. However, any

genes with a similar function to Bs-YjiC in *G. uralensis* are yet to be uncovered. Further identification and characterization of these UGTs are important to further our understanding of their roles in triterpenoid glycosylation, as well as in the regulation of the accumulation of GA-glycosylated derivates in *G. uralensis*.

In this work, we identified and characterized a new UGT named GuUGT from *G. uralensis*. Phylogenetic analysis suggested that this GuUGT may function on the 3-OH site of substrates. Heterologous expression in *Escherichia coli* was applied to investigate whether the recombinant protein functioned via glycosylation of GA. In addition, expression of GuUGT in *Saccharomyces cerevisiae* was performed to test its *in vivo* glycosylation of 3-OH and 30-COOH groups.

# 2. Material and methods

## 2.1. Plant materials and preparation

The seeds of *G. uralensis* Fisch were cultured in an artificial climate box controlled at 25°C with a 16 h : 8 h light–dark cycle. The roots, leaves and stems were collected from one-month-old plants, and a group was treated with 150 mM NaCl for 48 h for triterpenoid accumulation [16]. All samples were immediately frozen in liquid nitrogen and stored at −80°C.

## 2.2. Total RNA isolation and cDNA synthesis

Total RNA was extracted from the treated roots with a RNA prep Pure Plant Kit (Tian gen Biotech, Beijing, China) and then reverse-transcribed to generate cDNA with the use of the TransScript II cDNA Synthesis Supermix (Transgen Biotech, Beijing, China) in accordance with the manufacturer's protocol. According to previously published transcriptome data [16], the ORF (open reading frame) of GuUGT was amplified with La Taq (Takara, Dalian, China) using the PCR primers shown in electronic supplementary material, table S1 and then inserted into the pMD19T vector (pMD19T-GuUGT) for sequencing (Qingke Biotech, Beijing, China).

## 2.3. Tissue-specific expression analysis of *GuUGT* in *Glycyrrhiza uralensis*

The distribution of GuUGT transcripts in various tissues, including the root, leaf and stems, was examined by quantitative reverse-transcription PCR on an ABI PRISM 7500 real-time PCR system (Thermo Fisher Scientific, Wilmington, DE, USA) using a Fast SYBR™ Green Master Mix (Thermo Fisher Scientific, Wilmington, DE, USA). PCR was performed as follows: 3 min at 94°C, followed by 35 cycles of 94°C for 30 s, 62°C for 30 s and 72°C for 60 s. Primers used for the qRT-PCR analysis of GuUGT are listed in electronic supplementary material, table S2. The relative expression level was calculated according to the $2^{-\Delta\Delta CT}$ method. β-Actin (GenBank accession number EU190972.1) was used as the reference gene to normalize the expression of the candidate genes [23]. Three experimental replicates were performed for each gene. Statistical analysis was performed using SPSS 20.0 software (IBM SPSS Inc., Armonk, NY, USA). The ANOVA test was applied and significant differences were determined using Duncan's multiple range test with $p < 0.05$ indicating statistical significance.

## 2.4. Sequence alignment and phylogenetic analysis

The amino acid sequences of the candidate UGTs were aligned using ClustalW and MEGA 7.0.14 (https://www.megasoftware.net/) was used to construct a maximum likelihood bootstrapped phylogenetic tree. A neighbour-joining tree was constructed with the bootstrap method with 1000 replications.

## 2.5. Heterologous expression and purification from *Escherichia coli*

The ORF of GuUGT was cloned into expression vector pET32a(+) (addgene:11516), which inserted a 6×His-tag at the N-terminus of GuUGT. First, the pMD19T-GuUGT vector was digested with restriction enzymes KpnI and XhoI, and the GuUGT ORF sequence was ligated into a pET32a(+) vector that had been previously digested. The recombinant plasmid pET32a-GuUGT was transformed into *E. coli* Transetta (DE3) competent cells (Transgene Biotech, Beijing, China) according to the manufacturer's protocol. The positive transformants were inoculated into 10 ml Luria broth (LB) containing ampicillin (100 mg l$^{-1}$) at 37°C overnight. Then, 1% (1 ml/100 ml) of inoculum was added into 1 l fresh LB containing ampicillin

(100 mg l$^{-1}$) at 37°C until the OD$_{600}$ reached 0.5–0.7. Expression was induced with 0.2 mM IPTG and incubation at 16°C for 20 h with shaking at 180 r.p.m. The cells were harvested by centrifugation and then resuspended for lysis with a low-temperature ultra-high-pressure cell disrupter in lysis buffer (50 mM Tris-HCl pH 8.0, 150 mM NaCl, 10 mg ml$^{-1}$ lysozyme). After the supernatant was obtained by centrifugation at 13 000 r.p.m. for 35 min at 4°C, the recombinant protein partly soluble in the supernatant was purified by Ni-NTA affinity chromatography. The target bound protein was eluted from the Ni-NTA column with elution buffer (300 mM imidazole, 50 mM Tris-HCl pH 8.0) which was then exchanged with storage buffer (20 mM Tris-HCl pH 8.0, 150 mM NaCl) by centrifugation in 30 kDa cut-off ultracentrifugal filters. The purified recombinant protein was analysed by 10% SDS-PAGE, and its concentration was determined with a microplate reader using Coomassie Brilliant Blue G-250.

## 2.6. *In vitro* enzyme assays

Enzyme reactions were performed in 100 µl of buffer (50 mM Tris-HCl pH 8.0) containing 50 µg purified protein, 1 mM UDP-Glc (electronic supplementary material, figure S1, Yuanye Biotech, Shanghai, China) and 100 µM glycyrrhetinic acid (Sigma-Aldrich, Shanghai, China). After 6 h of incubation at 37°C, the reactions were stopped by the addition of 100 µl methanol. The conversion rates were calculated by HPLC peak area ($A_{product}/A_{substrate} + A_{product} \times 100\%$). The supernatants obtained by centrifugation (13 500 r.p.m. for 5 min) were analysed by Agilent 1260 HPLC (Agilent Technologies, Santa Clara, CA, USA) and LC-QTOF-MS (Agilent Technologies, Santa Clara, CA, USA), and internal standards (GA-3-O-monoglucose and GA-30-O-monoglucose) were supplied by Prof. Ye Min's laboratory (State Key Laboratory of Natural and Biomimetic Drugs, School of Pharmaceutical Sciences, Peking University).

## 2.7. Verification of the function of GuUGT by expression in *Saccharomyces cerevisiae* BY 4742

The key genes along the biosynthetic pathway for the production of GA and the expression of GuUGT include the genes encoding GuCYP88d6, GuCYP72a63, VvCPR, AtUDH and GuUGT. These genes were cloned together with restriction sites at their 5' and 3' ends into their corresponding plasmids. Detailed information regarding all the plasmids used in this study is provided in electronic supplementary material, table S3, and the primers used during plasmid construction are summarized in electronic supplementary material, table S4

Transformation of the *S. cerevisiae* strain was performed with the standard lithium acetate method. Strain BY-UGT was constructed by integrating the *Atudh* and GuUGT genes into the *EGH* sites of strain BY-β-AS, followed by selection on SD-LEU-URA-TRP plates. Strain BY-UGT-CYP was constructed by integrating *SvvCPR*, *CYP88D6* and *CYP72A63* into the *NDT80* site of strain BY-UGT followed by selection on SD-LEU-URA-TRP-HIS plates. All strains were verified by PCR analysis, and colonies were screened by GC-MS using an Agilent Technologies 5975C insert xl MSD with triple-axis detector equipped with a HP-5 ms (30 m × 0.25 mm × 0.5 µm) GC column (Agilent Technologies, Santa Clara, CA, USA) and LC-QTOF-MS (Agilent Technologies, Santa Clara, CA, USA) for the selection of transformants. The primers and all the strains used in this study are summarized in electronic supplementary material, tables S5 and S6, respectively.

All yeast strains, cultivated on the corresponding selection medium, were firstly inoculated into 15 ml culture tubes containing 2 ml synthetic dropout medium and grown at 30°C and 250 r.p.m. to an OD$_{600}$ of approximately 1.0. Flasks (100 ml) containing 10 ml synthetic dropout medium were then inoculated to an OD$_{600}$ of 0.05 with these seed cultures. Strains were grown at 30°C and 250 r.p.m. for 6 days [24].

For the extraction of the GA-glycoside derivates from the fermentation culture cells and the culture supernatant, different extraction buffers and methods were used. The cells were collected by centrifugation and added to 600 µl of an acetone : methanol mixture (1 : 1) and crushed three times with a Bead Beater (BioSpec, BS, USA). The samples were then centrifuged at 10 000$g$ for 1 min and the supernatant was analysed by GC-MS using an Agilent Technologies 5975C insert xl MSD with triple-axis detector equipped with a HP-5 ms (30 m × 0.25 mm × 0.5 µm) GC column (Agilent Technologies, Santa Clara, CA, USA) and LC-QTOF-MS (Agilent Technologies, Santa Clara, CA, USA). For the fermentation supernatant, 3 ml ethyl acetate was added to the supernatant followed by stirring in a shaker for 2 h at 30°C. Subsequently, the 1 ml upper organic phase was collected by centrifugation and evaporated with a rotary evaporator. The resultant compound was resuspended with 200 µl of 50% methanol in water and analysed by HPLC and LC-ESI-MS.

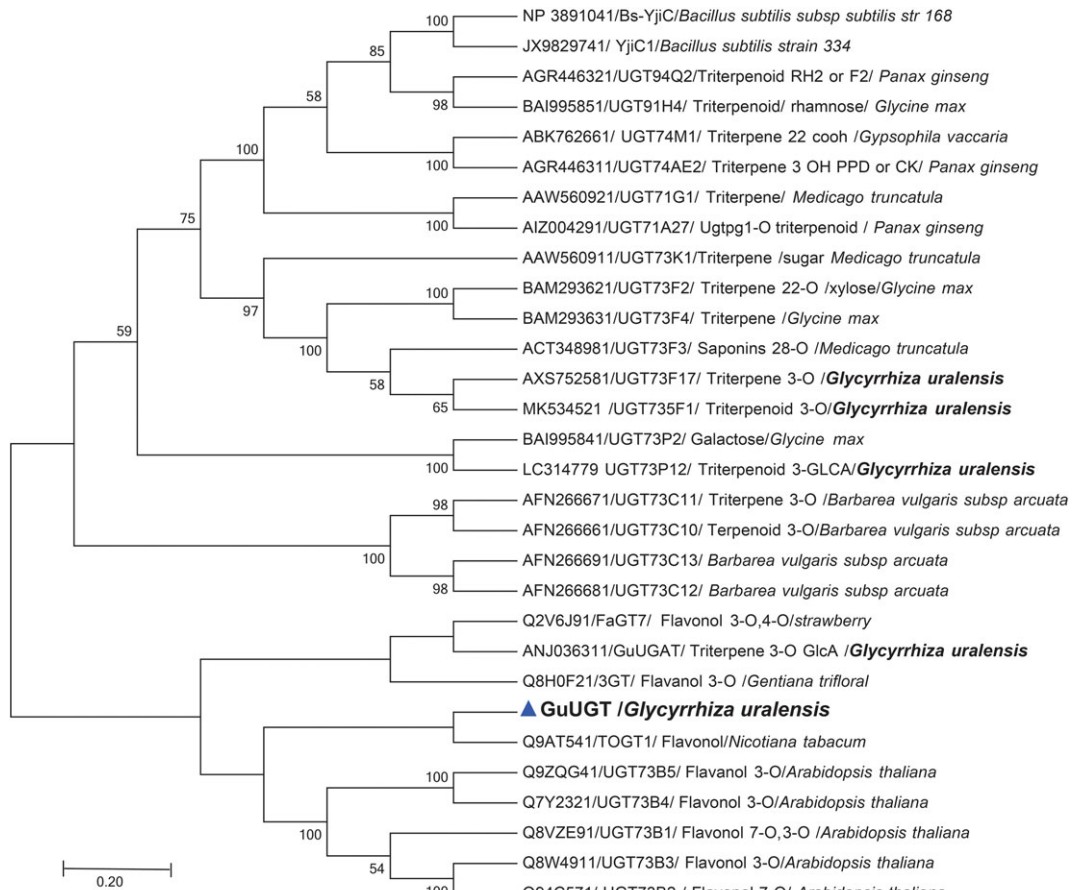

**Figure 2.** Phylogenetic relationships between UGTs of *G. uralensis* and other reported UGTs.

## 2.8. HPLC and LC-ESI-MS of products

HPLC analysis was carried out with an Agilent 1260 instrument. Chromatographic separation was performed on a reverse-phase C18 column (5 µm; 250 × 4.6 mm; Agilent Technologies) at room temperature with UV detection at 254 nm. MS operating conditions were as follows: all spectra were obtained in positive mode over an *m/z* range of 100–1200; dry gas flow, $6.0 \, l \, min^{-1}$; dry temperature, 180°C; nebulizer pressure, 1 bar; probe voltage +4.5 kV. A linear gradient elution method was employed with the mobile phase consisting of acetonitrile containing 0.1% formic acid (A) and 0.1% formic acid in water (B) at a flow rate of $1 \, ml \, min^{-1}$ using: A, 0–20 min at 35–95%; B, 20–35 min at 95–35%.

# 3. Results

## 3.1. cDNA cloning, sequence comparison and phylogenetic analysis

Based on previously published transcriptome datasets for *G. uralensis* [16], GuUGT was identified among 434 putative UGTs. To predict whether GuUGT is involved in the glycosylation of GA-glycosylate derivates, an unrooted neighbour-joining phylogenetic tree was constructed based on the sequences of full-length UGT proteins of functionally characterized triterpenoid and glycosylated-GA UGTs and UGTs identified as having more than 50% sequence identity with these UGTs as determined by Protein BLAST (https://blast.ncbi.nlm.nih.gov/Blast.cgi) searches against the Swiss-Prot database. The resultant phylogenetic tree suggested that these UGTs all probably function as glycosyltransferases of flavonols (figure 2). Alignment analysis revealed the presence of a conserved PSPG box at the C-terminus of all UGTs in this cluster (data not shown).

## 3.2. Enzyme activity and product identification

The full-length ORF of GuUGT was cloned into the expression vector pET32a and the recombinant enzyme was expressed with an N-terminal His-tag in *E. coli*. The recombinant GuUGT was then

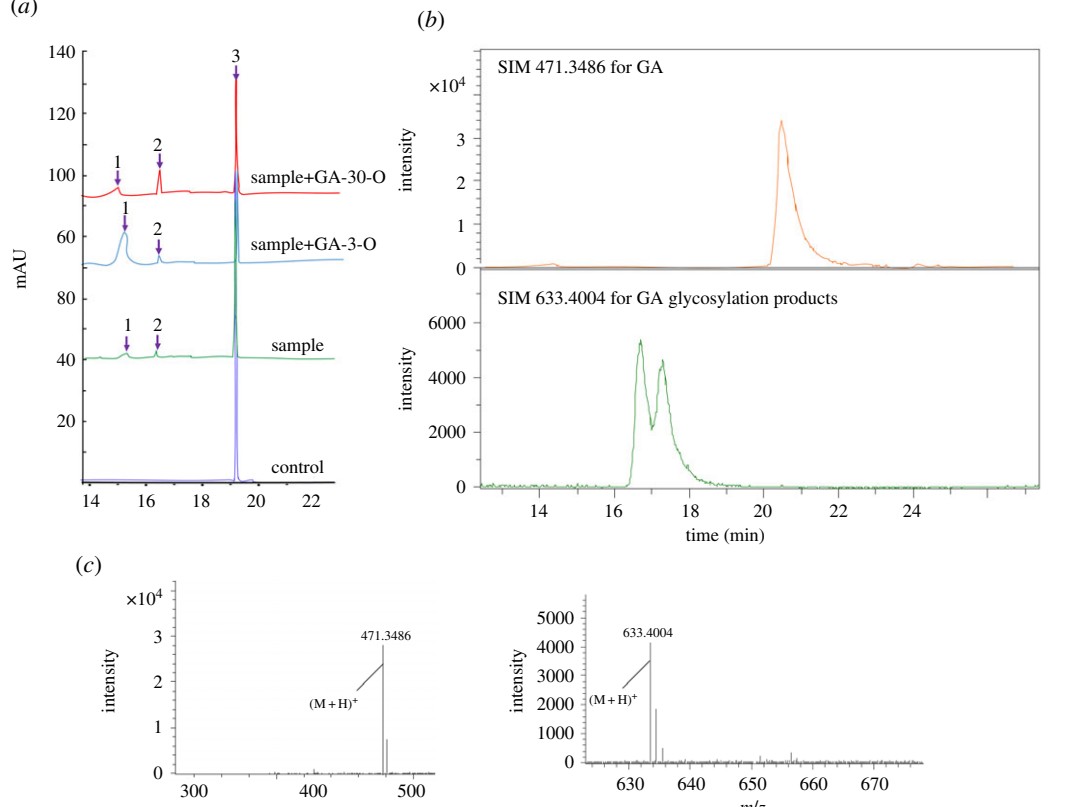

**Figure 3.** Glycosylation of GA by the purified glycosyltransferase (recombinant GuUGT). (*a*) HPLC analysis of the glycosylation products of GA catalysed by GuUGT; 1: GA-3-O-monoglucose; 2: GA-30-O-monoglucose; 3: GA. (*b*) SIM spectra of the glycosylation products of GA catalysed by GuUGT (GA was *m/z* 471.3486; GA glycosylation products was 633.4022). (*c*) MS spectra of glycosylation products: GA (*m/z* 471.3486), GA-3-O-monoglucose (*m/z* 633.4022), GA-30-O-monoglucose (*m/z* 633.4022).

purified by Ni-NTA chromatography and the presence of the target protein was confirmed by SDS-PAGE (electronic supplementary material, figure S2). The enzyme activity of GuUGT toward GA was investigated at pH 8.0 and 37°C. *In vitro* assays using UDP-glucose as the sugar donor indicated that the protein could catalyse glucosyl transfer to the 3-OH and 30-COOH sites of GA (figure 3), and 3-O-monoglucose GA ($[M+H]^+$ ion at *m/z* 633.4044) and 30-O-monoglucose GA ($[M+H]^+$ ion at *m/z* 633.4044) were detected as products. These products were verified using HPLC-Q-TOF-MS by comparing them with internal standards and the mass of the molecular ion. Higher conversion rate ($15.5\% \pm 2.1$) of 3-O-monoglucose GA than 30-O-monoglucose ($4.5\% \pm 1.6$) in the *in vitro* assay indicated that 3-O-monoglucose GA was the main product of GuUGT.

In addition, the sugar-donor specificities of the recombinant GuUGT were investigated, using UDP-glucose and UDP-glucuronide. The recombinant GuUGT was found to only accept UDP-glucose for the production of 3-O-monoglucose GA and 30-O-monoglucose GA (data not shown).

According to the result of phylogenetic analysis, we further tested flavonols, isorhamnetin as substrate, using UDP-glucose as the sugar donor. The results indicated GuUGT could accept isorhamnetin as substrate, and it produced isorhamnetin-3-O-glucoside at a considerably high conversion rate of $99.1\% \pm 3.1$. This product was confirmed by comparison with the authentic standard in terms of retention time (24.651 min) and exact mass of the protonated ion $[M+Na]^+$ ($m/z = 501.2013$) (electronic supplementary material, figure S3).

## 3.3. Function of GuUGT in yeast

We next investigated whether the catalytic function of GuUGT *in vivo* was consistent with the *in vitro* results. We chose yeast as an *in vivo* validation system for glucosyltransferase activity because it is a rapid and easy-to-use eukaryotic system with post-translational modifications that are similar to those of plants.

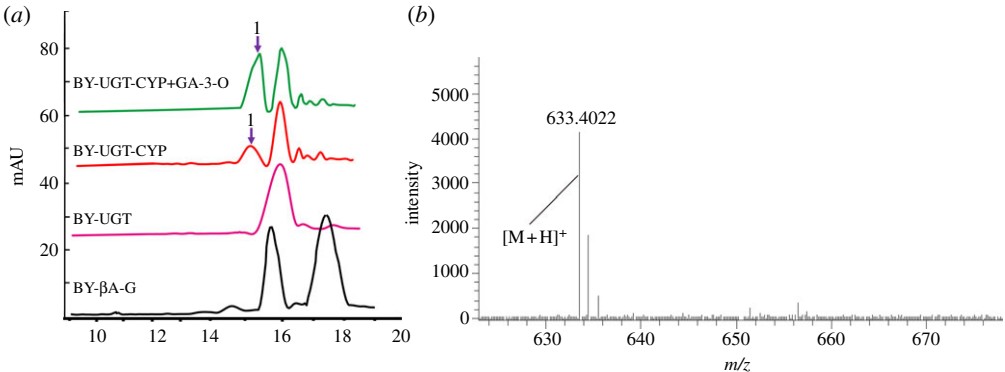

**Figure 4.** De novo glycosylated GA production by the yeast strain BY-UGT. (*a*) HPLC analysis of cell extracts from the various engineered yeast strains; 1: GA-3-O-monoglucose. (*b*) MS analysis of glycosylation products of GA from the yeast strain BY4742.

To characterize the function of GuUGT in yeast, we first introduced the plant P450 and P450 reductase genes into the BY-βA-G yeast strain to enable the biosynthesis of the substrate GA. However, the engineered yeast did not grow as expected, because GA showed antimicrobial activity that made it toxic to yeast. Thus, we chose to introduce GuUGT and Arabidopsis UDP-glucose dehydrogenase (*AtUDH*) together into the BY-βA-G strain for the construction of the BY-UGT strain, and *AtUDH* was used to catalyse the conversion of UDP-glucose within yeast to form UDP-GlcA. Finally, we introduced *SvvCPR*, *CYP72A63* and *GuCYP88D6* together in order to construct the BY-GA-glucoside derivate strain. To determine whether GuUGT enabled the synthesis of GA-glucoside derivates in yeast, we cultured the genetically manipulated yeast for 6 days. This engineered yeast strain produced new LC-ESI-MS peak at 15 min, which was identical to the retention time and molecular weight of 3-O-monoglucose GA([M + H]$^+$ ion at *m/z* 633.4044). Our results indicated that the GuUGT we identified was able to catalyse the conversion of GA *in vivo* in yeast (figure 4).

## 3.4. GuUGT expression in *Glycyrrhiza uralensis*

Since liquorice root has been traditionally used for medicinal purposes, we further examined in which parts of the plant GuUGT is mainly expressed. Transcript levels of GuUGT were determined by qRT-PCR using RNA extracted from the root, stem and leaves of *G. uralensis*. Transcript levels of GuUGT were detected in the root but were not detected in the stem and leaves of 15-day-old *G. uralensis* plants (figure 5).

# 4. Discussion

*Glycyrrhiza uralensis* triterpenoid saponins show a wide variety of biological activities. Their structural diversity is generated by sequential oxidations and, especially, by glycosylation. The *G. uralensis* genome has revealed as many as 434 annotated UGT genes; however, the biochemical functions of most of these remain to be clarified. Recently, four advanced studies characterized two UGT genes involved in triterpenoid saponin biosynthesis in *G. uralensis* [16–19]. Here, we identified and characterized another UGT, termed GuUGT. The expression level of GuUGT appears to be tissue-specific because GuUGT transcripts were only observed in the young root but not in the stem and leaves. This is consistent with the results of a previous study [16], and most triterpenoid saponins are extracted from the root of *G. uralensis* [7].

We performed phylogenetic analysis, which revealed that GuUGT clustered with glycosyltransferases that catalyse glucosylation at the 3-OH site, suggesting that it is homologous to enzymes capable of catalysing ether formation at C-3. This result is consistence with the result of GuUGT *in vitro* assay that the major products were identified as 3-O-glucosides. These enzymes are phylogenetically close to flavonoid rather than triterpenoid glycosyltransferases and are also supported by the higher conversion rate of GuUGT toward flavones than triterpenoids, when UDP-glucose was used as sugar donor. However, GuUGT is able to catalyse both flavones and triterpenoids as sugar acceptor substrates. Thus, phylogenetic analysis was not fully useful in this instance for determining the

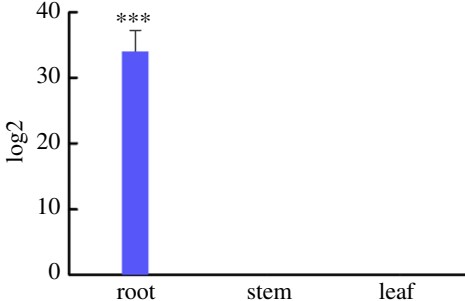

**Figure 5.** Expression of GuUGT transcripts in the leaf, stem and root of *G. uralensis* as determined by qRT-PCR. Different numbers of asterisks represent significant differences between groups (*$p < 0.05$, **$p < 0.01$, ***$p < 0.0001$).

biochemical properties of the identified enzyme. Such incongruence between phylogenetic position and substrate specificity has been observed for other UGTs. Strawberry FaGT7 was characterized as a flavonoid glucosyltransferase but clustered with triterpenoid glucosyltransferases [25]. It is notable that YjiC1 and BsYjiC from bacteria, UGT73C11 from *Barbarea* and GuUGT73F15 did not cluster with the same group as GuUGT, even though their functions have some similarities [19–22]. These results support the possibility that the biochemical characteristics of UGTs cannot always be accurately determined based on their protein sequences alone, and combined phylogenetic and experimental analysis is probably the most efficient approach for identifying UGT functions. Further work is required to explore the substrate promiscuity of GuUGT, including toward other flavone substrates with different structure, to determine whether the results are completely consistent with those of phylogenetic analysis.

The cloning of the ORF encoding GuUGT provided several insights into triterpenoid saponin biosynthesis in *G. uralensis*. The characterization of GuUGT products indicated that it is a triterpene glucosyltransferase that transfers a glucosyl moiety to the 3-OH of GA. In vitro, we found that the recombinant enzyme is capable of glycosylating GA at both 3-OH and 30-COOH, as determined by LC-ESI-MS. It is possible that different types of glycoside bonds were formed by UGT catalysis; for instance, ApUFGT2 can catalyse the formation of N- and O-glycoside bonds [26]. However, *in vivo* enzyme assays demonstrated that GuUGT prefers to catalyse the transfer of glucose to the 3-OH site of GA to produce 18β-GA-3-*O*-β-D-glucoside. Similarly, citrus CsUGT76F1 has been shown to glycosylate the 7-OH group of flavonoids *in vivo* but glycosylate 3-OH and 7-OH *in vitro* [27]. $K_{cat}K_m^{-1}$ values and metabolite profiling of MtUGT71G1 also suggested *in vitro* and *in vivo* differences in the preferred substrate of UGT71G1 [28]. This phenomenon may be attributable to the complicated reaction conditions present *in vivo*. Another possible factor determining enzyme activity may be the different relative concentrations of the sugar donor.

In this study, we reported the discovery of GuUGT from *G. uralensis*, which is a glucosyltransferase that uses a glucosyl moiety as the sugar donor. Phylogenetic analysis showed that GuUGT appears to be related to glucosyltransferases capable of catalysing transfer to the 3-O position of flavanols rather than triterpenoids. In contrast, biochemical analysis *in vitro* revealed that recombinant GuUGT successfully glycosylated GA to form 3-O-monoglucose or 30-O-monoglucose. GuUGT activity *in vivo* in yeast conferred biosynthesis of 18β-GA-3-*O*-β-D-glucoside. In the future, we hope to investigate the substrate promiscuity of GuUGT and further explore its biological role in plants.

**Data accessibility.** The coding sequence (CDS) of GuUGT and electronic supplementary material from this study were available online at https://figshare.com/s/317bcf409ab6e39b2812.

**Authors' contributions.** Y.H. carried out the molecular cloning work. D.L. participated in data analysis. J.W. prepared the experimental material. Y.C. carried out LC-MS and GC-MS analysis. Z.D. participated in the design of the study and drafted the manuscript. D.J. and C.L. conceived of the study, designed the study, coordinated the study and helped draft the manuscript. All authors gave final approval for publication.

**Competing interests.** The authors declare that they have no conflicts of interest.

**Funding.** This research did not receive any specific grant from funding agencies in the public, commercial or not-for-profit sectors.

**Acknowledgements.** We thank Prof. Ye Min for providing the GA-3-*O*-D-glucoside and GA-30-*O*-D-glucoside. We thank Natasha Beeton-Kempen, PhD, from Liwen Bianji, Edanz Editing China (www.liwenbianji.cn/ac), for editing the English text of a draft of this manuscript.

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
