## [Reviewer comments · Royal Society Open Science]

Review History

RSOS-191121.R0 (Original submission)

Review form: Reviewer 1

Is the manuscript scientifically sound in its present form?

No

Are the interpretations and conclusions justified by the results?

No

Is the language acceptable?

Yes

Do you have any ethical concerns with this paper?

No

Have you any concerns about statistical analyses in this paper?

Yes

Recommendation?

Major revision is needed (please make suggestions in comments)

Comments to the Author(s)

In this manuscript the authors have described the characterisation of a UGT from *Glycyrrhiza* which contributes toward the understanding of the substrate specificity of UGTs in general.

There are some revisions that need to be made for this manuscript is ready for publication.

Line 1 The use of the term 'permissive' is misleading as the authors have only demonstrated that GuUGT glucosylates one molecule at two positions. It should be removed from the title.

Line 16 'glucosyltransferases' should be replaced with 'glycosyltransferases' as GuUGAT is a glucuronosyltransferase not a glucosyltransferases.

Line 59 Should refer to 'GuUGAT' not 'GuUGT'.

Line 61 Reference required for GuUGAT.

Line 106 Reference required for method (see comments about Figure 6 below)

Lines 155, line 235 GuCYP88d6 and CuCYP72a63 are written inconsistently throughout the manuscript. The subgroup needs to be in capital letters ie GuCYP88D8 and GuCYP72A63.

Line 190 Need to add the mass spec conditions.

Line 232 Why was AtUDH introduced into the yeast strain? GuUGT uses UDP-glucose so why was it necessary to produce UDP-GlcA?

Line 244 Replace full stop with comma.

Line 264 '.... not so closely.' This sentence does not make sense and needs to be rewritten.

Line 282 Could the reason why 3-OH rather than 30-COOH is the preferred position in vivo be due to instability of the glucose ester bond?

Table1, Table 2, Figure 3 should be in supplementary data.

Figure 1 is poorly drawn. Why do the text sizes vary? Why are some of the enzymes in boxes with black backgrounds but others aren't? Why is CYP72A154 referred to in this figure rather than CYP72A63 which was stated to be one of the key genes in line 155?

Figure 4A What do 1,2 and 3 refer to? They aren't mentioned in the figure legend. Axis also need labelling.

Figure 4B This is too crowded and it is difficult to work out what is going on. Please separate out the overlaid images and write a more details legend to explain what is being referred to.

Figure 5A Axis labels are required. What does 1 refer to?

Figure 6 I am unable to understand from the methods or the results what the expression levels in the roots is relative to. Relative expression level requires a control sample to act as a comparison. Please explain how this was calculated because this data can't be published as it is at the moment. Is the relative expression level n-fold or log2?

Review form: Reviewer 2

Is the manuscript scientifically sound in its present form?

No

Are the interpretations and conclusions justified by the results?

Yes

Is the language acceptable?

Yes

Do you have any ethical concerns with this paper?

No

Have you any concerns about statistical analyses in this paper?

No

Recommendation?

Major revision is needed (please make suggestions in comments)

Comments to the Author(s)

Huang et al. discovered a permissive GT from *Glycyrrhiza uralensis* which could catalyze the 3- and 30- O-glycosylation of glycyrrhetic acid. It might be considered for publication in Royal Society Open Science after a major revision.

1. Several GTs had been reported from *Glycyrrhiza* species recently (The Plant journal, DOI: 10.1111/tpj.14409; ACS Synthetic Biology, 10.1021/acssynbio.9b00171). These GTs needs to be included in Figure 2, and the difference between GuUGT and these enzymes needs to be discussed.

2. The yield or conversion rate needs to be clearly stated. According to Figure 4A, the yield of O-glycosides is poor. Is it possible that triterpenes are not its favorite substrate? Did the authors try other types of substrates?

3. The chromatographic peak shapes were poor in Figure 4 and Figure 5, and the retention times were different for HPLC/UV and HPLC/MS analysis. Please explain.

4. What is the product difference between the reactions catalyzed by recombined enzyme (Figure 4B) and yeast (Figure 5B)? Was GA-30-O-Glu detected in yeast?

5. Minor points.

Line 186: reverse-phase C18 column (5 mL; 250×4.6 mm; Agilent Technologies). Should it be 5 μ m?

Line 51: 18 β -GA is one of the most important oleanane-type triterpenoid saponins in *G. uralensis*. GA is a triterpenoid sapogenin, not a saponin.

Line 190: A, 0-20 min at 35-95%; B, 20-35 min at 95-35%. A was water and B was acetonitrile. Is the elution gradient correct when a reverse-phase column was used?

Review form: Reviewer 3

Is the manuscript scientifically sound in its present form?

Yes

Are the interpretations and conclusions justified by the results?

Yes

Is the language acceptable?

Yes

Do you have any ethical concerns with this paper?

No

Have you any concerns about statistical analyses in this paper?

No

Recommendation?

Accept with minor revision (please list in comments)

Comments to the Author(s)

In the manuscript, the authors isolated a UGT with expression in the roots of *G. uralensis* and biochemical analyses revealed that the recombinant UGT enzyme could transfer a glucosyl moiety to the free 3-OH or 30-COOH groups of GA. In addition, the authors were also engineered yeast harbouring genes involved in the biosynthetic pathway for GA-glycoside derivatives produced GA-3-O- β -D-glucoside, implying that the enzyme has GA 3-O-glucosyltransferase activity *in vivo*. The study is solid and the results reported are interesting. The manuscript is also nicely written. Therefore, this manuscript is worth publication. Following please a few examples to improve this manuscript.

- 1) Line 17 The "UGT" should not be italic .
- 2) Line 70 "a new UGT from..." should be a new UGT named GuUGT
- 3) Line 114 MEGA 7.0 should added the manufacturer of the software .
- 4 Line 153 What kind of *Saccharomyces cerevisiae* was the author use ? And they should supplied.
- 6) Line 142 The chemical structure of UDP-Glc may supplied in supplemental materials,
- 7) Line 166 The manufacturer of GC/MS and LC-ESI-MS please correct all in throughout the whole manuscript.
- 8) The legend of Figure 2 the "various other reported UGTs" should be revised as other reported UGTs.
- 9) The SEQUENCE Table 1 and table 2 should be lowercase
- 10) In introduction and discussion part, some recent references about UGT could be added to broaden the discussion depth (Functional characterization of three flavonoid glycosyltransferases from *Andrographis paniculata* in Royal Society Open Science 6(6):190150; Organic letters, 2018, 20(19): 5999-6002)

Decision letter (RSOS-191121.R0)

05-Aug-2019

Dear Dr Huang,

The editors assigned to your paper ("GuUGT, a permissive glycosyltransferase from *Glycyrrhiza uralensis*, exhibits glycyrrhetic acid 3- and 30-O-glycosylation") have now received comments from reviewers. We would like you to revise your paper in accordance with the referee and Associate Editor suggestions which can be found below (not including confidential reports to the Editor). Please note this decision does not guarantee eventual acceptance.

Please submit a copy of your revised paper before 28-Aug-2019. Please note that the revision deadline will expire at 00.00am on this date. If we do not hear from you within this time then it will be assumed that the paper has been withdrawn. In exceptional circumstances, extensions may be possible if agreed with the Editorial Office in advance. We do not allow multiple rounds of revision so we urge you to make every effort to fully address all of the comments at this stage. If deemed necessary by the Editors, your manuscript will be sent back to one or more of the original reviewers for assessment. If the original reviewers are not available, we may invite new reviewers.

- Data accessibility

If you wish to submit your supporting data or code to Dryad (<http://datadryad.org/>), or modify your current submission to dryad, please use the following link:
<http://datadryad.org/submit?journalID=RSOS&manu=RSOS-191121>

- Competing interests

- Authors' contributions

- Acknowledgements

- Funding statement

on behalf of Professor Kalle Gehring (Associate Editor) and Andrew Dunn (Subject Editor)
openscience@royalsociety.org

Reviewers' Comments to Author:

Reviewer: 1

Comments to the Author(s)

In the manuscript, the authors isolated a UGT with expression in the roots of *G. uralensis* and biochemical analyses revealed that the recombinant UGT enzyme could transfer a glucosyl moiety to the free 3-OH or 30-COOH groups of GA. In addition, the authors were also engineered yeast harbouring genes involved in the biosynthetic pathway for GA-glycoside derivatives produced GA-3-O- β -D-glucoside, implying that the enzyme has GA 3-O-glucosyltransferase activity *in vivo*. The study is solid and the results reported are interesting. The manuscript is also nicely written. Therefore, this manuscript is worth publication. Following please a few examples to improve this manuscript.

- 1) Line 17 The "UGT" should not be italic .
- 2) Line 70 "a new UGT from..." should be a new UGT named GuUGT
- 3) Line 114 MEGA 7.0 should added the manufacturer of the software .
- 4) Line 153 What kind of *Saccharomyces cerevisiae* was the author use ? And they should supplied.
- 6) Line 142 The chemical structure of UDP-Glc may supplied in supplemental materials,
- 7) Line 166 The manufacturer of GC/MS and LC-ESI-MS please correct all in throughout the whole manuscript.
- 8) The legend of Figure 2 the "various other reported UGTs" should be revised as other reported UGTs.
- 9) The SEQUENCE Table 1 and table 2 should be lowercase

10) In introduction and discussion part, some recent references about UGT could be added to broaden the discussion depth (Functional characterization of three flavonoid glycosyltransferases from *Andrographis paniculata* in Royal Society Open Science 6(6):190150; Organic letters, 2018, 20(19): 5999-6002)

Reviewer: 2

Comments to the Author(s)

Huang et al. discovered a permissive GT from *Glycyrrhiza uralensis* which could catalyze the 3- and 30- O-glycosylation of glycyrrhetic acid. It might be considered for publication in Royal Society Open Science after a major revision.

1. Several GTs had been reported from *Glycyrrhiza* species recently (The Plant journal, DOI: 10.1111/tpj.14409; ACS Synthetic Biology, 10.1021/acssynbio.9b00171). These GTs needs to be included in Figure 2, and the difference between GuUGT and these enzymes needs to be discussed.

2. The yield or conversion rate needs to be clearly stated. According to Figure 4A, the yield of O-glycosides is poor. Is it possible that triterpenes are not its favorite substrate? Did the authors try other types of substrates?

3. The chromatographic peak shapes were poor in Figure 4 and Figure 5, and the retention times were different for HPLC/UV and HPLC/MS analysis. Please explain.

4. What is the product difference between the reactions catalyzed by recombined enzyme (Figure 4B) and yeast (Figure 5B)? Was GA-30-O-Glu detected in yeast?

5. Minor points.

Line 186: reverse-phase C18 column (5 mL; 250×4.6 mm; Agilent Technologies). Should it be 5 μ m?

Line 51: 18 β -GA is one of the most important oleanane-type triterpenoid saponins in *G. uralensis*. GA is a triterpenoid sapogenin, not a saponin.

Line 190: A, 0-20 min at 35-95%; B, 20-35 min at 95-35%. A was water and B was acetonitrile. Is the elution gradient correct when a reverse-phase column was used?

Reviewer: 3

Comments to the Author(s)

In this manuscript the authors have described the characterisation of a UGT from *Glycyrrhiza* which contributes toward the understanding of the substrate specificity of UGTs in general. There are some revisions that need to be made for this manuscript is ready for publication.

Line 1 The use of the term 'permissive' is misleading as the authors have only demonstrated that GuUGT glucosylates one molecule at two positions. It should be removed from the title.

Line 16 'glycosyltransferases' should be replaced with 'glycosyltransferases' as GuUGAT is a glucuronosyltransferase not a glucosyltransferases.

Line 59 Should refer to 'GuUGAT' not 'GuUGT'.

Line 61 Reference required for GuUGAT.

Line 106 Reference required for method (see comments about Figure 6 below)

Lines 155, line 235 GuCYP88d6 and CuCYP72a63 are written inconsistently throughout the manuscript. The subgproun needs to be in capital letters ie GuCYP88D8 and GuCYP72A63.

Line 190 Need to add the mass spec conditions.

Line 232 Why was AtUDH introduced into the yeast strain? GuUGT uses UDP-glucose so why was it necessary to produce UDP-GlcA?

Line 244 Replace full stop with comma.

Line 264 '.... not so closely.' This sentence does not make sense and needs to be rewritten.

Line 282 Could the reason why 3-OH rather than 30-COOH is the preferred position in vivo be due to instability of the glucose ester bond?

Table1, Table 2, Figure 3 should be in supplementary data.

Figure 1 is poorly drawn. Why do the text sizes vary? Why are some of the enzymes in boxes with black backgrounds but others aren't? Why is CYP72A154 referred to in this figure rather than CYP72A63 which was stated to be one of the key genes in line 155?

Figure 4A What do 1,2 and 3 refer to? They aren't mentioned in the figure legend. Axis also need labelling.

Figure 4B This is too crowded and it is difficult to work out what is going on. Please separate out the overlaid images and write a more details legend to explain what is being referred to.

Figure 5A Axis labels are required. What does 1 refer to?

Figure 6 I am unable to understand from the methods or the results what the expression levels in the roots is relative to. Relative expression level requires a control sample to act as a comparison. Please explain how this was calculated because this data can't be published as it is at the moment. Is the relative expression level n-fold or log2?

Author's Response to Decision Letter for (RSOS-191121.R0)

See Appendix A.

Decision letter (RSOS-191121.R1)

08-Sep-2019

Dear Dr Huang,

I am pleased to inform you that your manuscript entitled "GuUGT, a glycosyltransferase from *Glycyrrhiza uralensis*, exhibits glycyrrhetic acid 3- and 30-O-glycosylation" is now accepted for publication in Royal Society Open Science.

Please note that the email address max_liucs@263.com for Chunsheng Liu is not currently

receiving emails - please ensure that you not only forward this message on to Dr Liu but also provide us with an alternative email address for Dr Liu.

on behalf of Professor Kalle Gehring (Associate Editor)
openscience@royalsociety.org

Appendix A

Dear Editor:

Thank you very much for your supervision of the reviewing process of my manuscript (RSOS-191121). We must thank you and all other reviewers for the critical feedback. We feel lucky that our manuscript went to these reviewers as the valuable comments from them not only helped us with the improvement of our manuscript, but suggested some neat ideas for future studies. According to the reviewer's instructions, we have made the following revisions on this manuscript.

1. RESPONSES TO REVIEWER 1 S' COMMENTS

Thank you very much for your kindly comments on our manuscript . Based on your suggestions, we carefully revised the manuscript.

- 1) Line 17 The sentence "UGT" should not be italic. And we have revised this.
- 2) Line 70 "a new UGT from..." should be a new UGT named GuUGT.
- 3) Line 114 The website of MEGA 7.0 was <https://www.megasoftware.net/>, and we add this in our manscript.
- 4) *Saccharomyces cerevisiae* BY4742 was used in our manuscript.
- 6) The chemical structure of UDP-Glc was added in Figure S1
- 7) Line 166 We added the manufacturer of GC-MS and LC-ESI-MS in our whole manscript.
- 8) We have corrected the sentence.
- 9) We revised the Sequence in table 1 and table 2.
- 10) We have **cited** newly literatures in the introduction and discussion part. And we discuss the difference between GuUGT **and these UGTs**.

2. RESPONSES TO REVIEWER 2 S' COMMENTS

Thank you very much for your kindly comments on our manuscript.

1. We have added the UGT73P12 and UGT75F15 in Figure 2, and in the introduction discussion part we have discussed the difference between GuUGT and the two enzymes.

Thanks for your advice.

2. Yes, the conversion rate was poor of GuUGT, and we have added the conversion rate of GA-3-O- β -D-glucoside ($15.5\% \pm 2.1$, 6 hours), GA-30-O- β -D-glucoside ($4.5\% \pm 1.6$, 6 hours) to our manuscript. Meanwhile we did the additional experiment for explain why the conversion rate was so poor, and we use the isorhamnetin (flavonoids compounds) as substrate, and we found that the conversion rate was 99.1 ± 3.1 (2 hours) and we have added HPLC-MS analysis of the glycosylation products of isorhamnetin (Figure S3).

Ps. We used Shimadzu LC/MS 8045 to finish this experiment.

3. We have consult the operator of LC-qTOF-MS for the retention delay between HPLC/UV and HPLC/MS, due to the HPLC connect two instruments include NMR and qTOF MS, and liquid phase pipeline is longer, resulted in the retention times were different for HPLC/UV and HPLC/MS analysis. And the photo of the instruments was below.

4. In the MS TIC ion flow, the products in the reactions catalyzed by recombined enzyme (figure 4 B) was GA-3-O- β -D-glucoside and GA-30-O- β -D-glucoside, and the products in the reactions catalyzed by yeast was GA-3-O- β -D-glucoside (figure 5B). We didn't detect the GA-30-O- β -D-glucoside in the yeast, and we think the reason may due to the glycosylation ability of GuUGT was differently displayed *in vitro* and yeast. Other UGTs such as crocin UGTs have the glycosylation ability *in vitro* and shown no catalytic activity *in vivo*. Other reason might be inadequate supply of precursor of GA in yeast resulting in the plant-derived glycosyltransferases preferentially catalyzing position of 3-OH of GA in yeast.

5.

Line 186: We are sorry to make this mistake, and we have corrected the 5 mL to 5 μ m.

Special thanks to you for your good comments.

Line 51: We have revised the sapogenin in line 51.

line 190: Thanks for your advice and we are sorry for our negligence of the elution gradient, and we have revised this: A, 0-20 min at 35-95%; B, 20-35 min at 95-35%. A was acetonitrile and B was water.

3. RESPONSES TO REVIEWER 3 S' COMMENTS

We are truly grateful to your critical comments and thoughtful suggestions. Based on these comments and suggestions, we have carefully revised the manuscript.

Line 1 we have removed the permissive from the title.

Line 16 we have replaced the glucosyltransferases with 'glycosyltransferases'.

Line 59 we have revised the GuUGAT.

line 61 we have added the reference of GuUGAT.

line 106 we have added the reference of method.

Line 155,235 We have revised the capital letters of GuCYP.

Line 190 We have added the mass spectra conditions in our manuscript.

Line 232 We have tried the UDP-GLCA and GA as substrate *in vitro*, unfortunately GuUGT could not catalyze UDP-GLCA, so we want to test if GuUGT can catalyze UDP-GLCA and GA in yeast, and we detected that there were no catalysate in the yeast.

Line 244 We have revised this.

Line 264 We have rewrite this sentence.

Line 282 We think the reason GuUGT in yeast that 3-OH rather than 30-COOH is the preferred position may due to the yield of GA in yeast is very low, resulting in GuUGT preferred 3-OH position.

We have redrawn the figure 1, and the enzymes in boxes with black backgrounds stands for the source of enzymes from plants or other organisms were heterologously transformed into yeast and we have removed these for keeping it simple. We have revised the CYP72A63, because the Prof. Li Chun reported that the enzyme activity of CYP72A63 from *Medicago truncatula* was higher than GuCYP72A154 in yeast.

(reference: Zhu M, Wang,C, Sun W, et al. 2018 Boosting 11-oxo- β -amyrin and glycyrrhetic acid synthesis in *Saccharomyces cerevisiae* via pairing novel oxidation and reduction system from legume plants.*Metabolic Engineering* 45 (2018) 43-50.(doi: doi.org/10.1016/j.ymben.2017.11.009))

Figure 4A, we have added the 1 2 3 to the figure 1 legends. And we have added the axis labels in Figure 4A. Axis labels were also added.

We have revised the Figure 4B, and added the MS spectra of glycosylation products in Figure 4C.

Figure 5A We have added the 1 to the figure legend, and 1 was refer to GA-3-O- β -D-glucoside. Axis labels were also added.

We feel sorry to make a mistake in Figure 6 and the Relative expression level actual was Log 2, and we use the β -Actin (GenBank accession number EU190972.1) was used as the reference gene, the final value were calculated according to the $2^{-\Delta\Delta CT}$ method. And in the steam and leaf group we didn't detected the gene expression of GuUGT.

Based on the comments we received, careful modifications have been made to the R1 manuscript. All changes were marked in red text. We hope that these revisions are satisfactory and that the revised version will be acceptable for publication in RSOS.

Thank you very much for your work concerning my paper.

Wish you all the best!

Sincerely yours